# Photon upconversion crystals doped bacterial cellulose composite films as recyclable photonic bioplastics
Pankaj Bharmoria [1] ✉, Lukas Naimovičius [1,2], Deyaa Abol-Fotouh [1,3], Mila Miroshnichenko[1], Justas Lekavičius [2], Gabriele De Luca [1], Umair Saeed[4], Karolis Kazlauskas[2], Nicolas Candau [5], Paulius Baronas[6], Anna Roig [1] & Kasper Moth-Poulsen [1,6,7,8] ✉

Biopolymers currently utilized as substitutes for synthetic polymers in photonics applications are predominantly confined to linear optical color responses. Herein we expand their applications in non-linear optics by integrating with triplet-triplet annihilation photon upconversion crystals. A photon upconverting biomaterial is prepared by cultivating Pd(II) meso-tetraphenyl tetrabenzoporphine: 9,10-diphenyl anthracene (sensitizer: annihilator) crystals on bacterial cellulose hydrogel that serves both as host and template for the crystallization of photon upconversion chromophores. Coating with gelatin improves the material's optical transparency by adjusting the refractive indices. The prepared material shows an upconversion of 633 nm red light to 443 nm blue light, indicated by quadratic to linear dependence on excitation power density (non-linearly). Notably, components of this material are physically dis-assembled to retrieve 66 ± 1% of annihilator, at the end of life. Whereas, the residual clean biomass is subjected to biodegradation, showcasing the sustainability of the developed photonics material.

Synthetic thermoplastics (PVA, PMMA, polystyrene, etc.) are rapidly taking over glasses in the photonics market due to their cost-effective bulk synthesis, easy molding and processing, and excellent thermo-mechanical and opto-electronic properties[1]. This transition is projected to drive the global photonics market to an estimated value of around 1300 billion USD by 2028[2]. However, the increased use of synthetic plastic raises concerns about environmental impact, as the disposal of non-biodegradable plastic waste from these photonic components at their end-of-life (EoL)[3], could lead to the accumulation of up to 12,000 million tonnes in landfills or natural environment by 2050, particularly in developing nations[4]. While thermo-plastics can be recycled through mechanical or chemical methods for a limited number of cycles, their non-biodegradable nature poses a significant challenge to long-term ecological sustainability[5,6]. As a result, there is a growing interest in utilizing biodegradable biopolymers, which, in some cases, exhibit competitive thermo-mechanical properties and offer advantages such as natural abundance and biodegradability. Although

biopolymers have been utilized in photonics applications, the research primarily focuses on generating linear optical color responses by diffraction or reflection of light by ordered or chiral assemblies of biopolymer nanostructures[7]. However, their bio-sustainability concerning the recovery of doped toxic photonic chromophores in those nanostructures at the EoL remains a key challenge for the future circular bioeconomy. We envision that one potential solution to address this challenge is pre-programmed co-assembly↔disassembly of chromophores crystals within a solid biopolymer matrix to facilitate physical recycling at the EoL.

In pursuit of this objective a simple pre-programmable method of "photon upconversion crystals doped biopolymers" has been devised. This method involves the crystallization of non-linear optically responsive photonic chromophores within a composite film of bacterial cellulose–gelatin (BC–G), as illustrated in Fig. 1b. The photonic crystals entrapped inside the films exhibit upconverted blue emission when excited with red light, and display quadratic to linear relationship with the excitation

[1]Institute of Materials Science of Barcelona, ICMAB-CSIC, Barcelona, Spain. [2]Institute of Photonics and Nanotechnology, Vilnius University, Vilnius, Lithuania. [3]Advanced Technology and New Materials Research Institute (ATNMRI), City of Scientific Research and Technological Applications (SRTA-City), New Borg Al-Arab, Egypt. [4]Catalan Institute of Nanoscience and Nanotechnology (ICN2), Barcelona, Spain. [5]Departament de Ciència i Enginyeria de Materials (CEM), Escola d'Enginyeria Barcelona-Est (EEBE), Universitat Politècnica de Catalunya BarcelonaTech (UPC), Barcelona, Spain. [6]Department of Chemical Engineering Universitat Politècnica de Catalunya, EEBE, Barcelona, Spain. [7]Catalan Institution for Research & Advanced Studies, ICREA, Barcelona, Spain. [8]Department of Chemistry and Chemical Engineering, Chalmers University of Technology, Gothenburg, Sweden. ✉e-mail: pbharmoria@icmab.es; kasper.moth-poulsen@chalmers.se

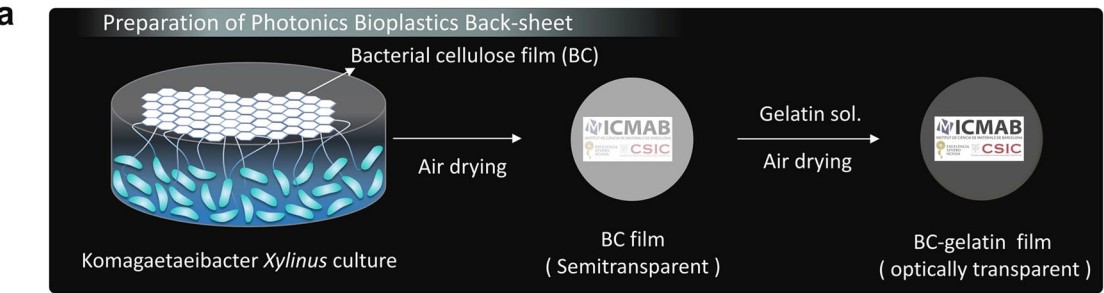

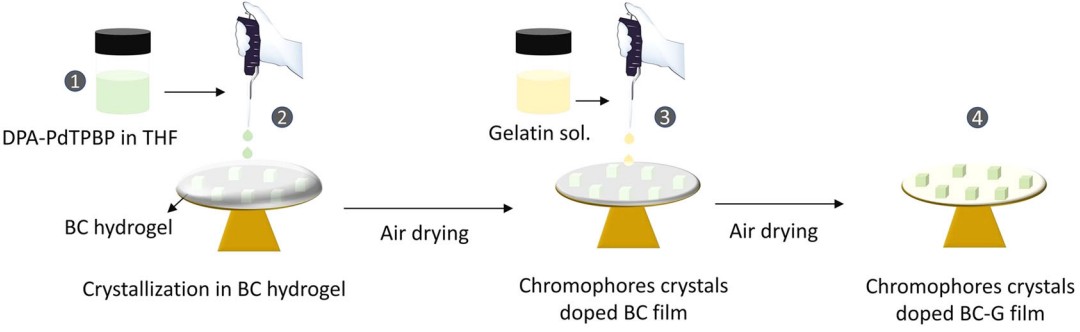

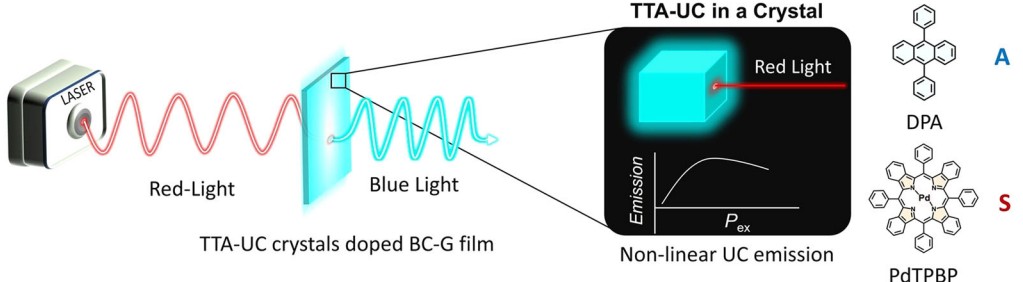

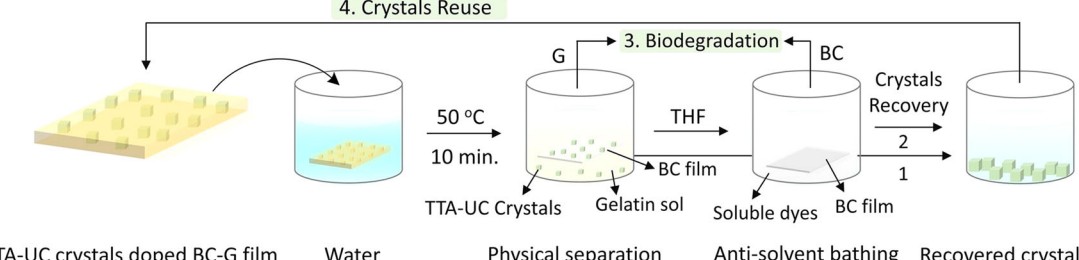

**Fig. 1 | Illustration of step-wise preparation of physically recyclable non-linear optically active photonics bioplastics. a** Preparation of transparent BC–G films as photonics back sheet. **b** Crystallization of DPA–PdTPBP TTA-UC crystals on BC–G film. **c** Light manipulation via photon upconversion by doped TTA-UC crystals in the film and molecular structures of annihilator (A) DPA and sensitizer (S) PdTPBP. **d** Recycling TTA-UC chromophores crystals via physical separation in water and anti-solvent bathing using THF.

power density (Fig. 1c). Interestingly, the doped chromophores crystals and BC film could be recycled through physical separation in water at the EoL, as depicted in Fig. 1d. The photon upconversion emission in crystals occurs via triplet–triplet annihilation photon upconversion (TTA-UC)[8] in air due to Dexter energy migration[9,10] within the closely packed crystals of 9,10-diphenylanthracene (DPA, annihilator) and Palladium(II) meso-tetraphenyl tetrabenzoporphine (PdTPBP, sensitizer)[11]. Following

excitation with low-energy red light, the sensitizer transfers its triplet energy to the dark triplets of the annihilator, which, upon sensitization, undergo annihilation due to triplet-exchange coupling, leading to the generation of an excited singlet state of the annihilator, emitting light at higher energy compared to excitation energy[11].

Solid-state TTA-UC materials offer the possibility of improving the efficiency of solar-to-electric conversion in photovoltaic (PV) and solar

**Fig. 2 | Comparative structure and properties of G, BC, and BC–G films.** Transmission electron microscopy images of **a** BC showing porous fibrous structure (inset translucent digital image of the film), and **b** BC–G composite film (inset transparent digital image of the film). **c** P-XRD spectra of the BC, G, and BC–G films. **d** Frequency dependence of the dielectric permittivity of BC films with different thicknesses, G film, and BC–G film.

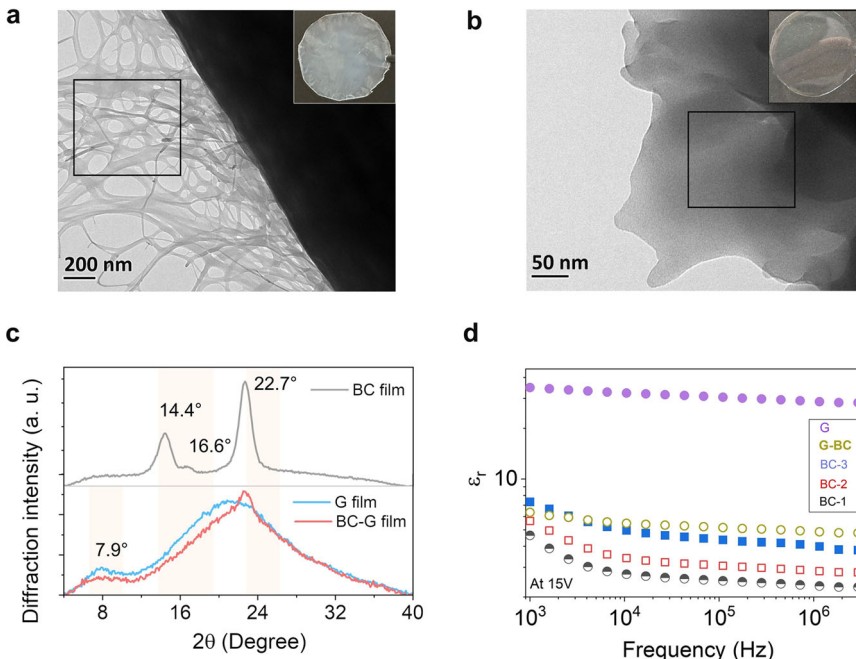

energy storage in molecular solar thermal energy storage (MOST) systems[12–14]. They can do it by increasing the photon flux at energies higher than the band gap of the active layer by upconverting the transmitted photons below the band gap[14]. However, one key issue of effectively capturing UC photons by PV/MOST system is their scattered emission. Recently, we reviewed sustainable approaches for TTA-UC integrated solar energy systems and found that crystalline TTA-UC system's inside the PV material can improve UC photon absorption in addition to providing a bed for better crystal packing of PV materials[14]. Additionally, to address the issue of plastic waste of solar panel (78 million tons of solar panel waste expected by 2050)[15] it is desired to fabricate TTA-UC integrated PV materials on biodegradable bioplastics for use as back-sheet materials.

Therefore, this work takes an important step in this direction by using BC–G composite film as back-sheet materials to integrate TTA-UC crystals. The validation of this composite film as a back-sheet material has been tested by dielectric, and thermo-mechanical analysis. Finally, recycling and reclaiming the toxic photonic chromophores is completed to complete the circular bioeconomy loop[4,16]. Since our process is safe by design including its recyclability, the extraction of chromophores was easily done by physical separation of the chromophore's crystals from the film in hot water at 50 °C, followed by anti-solvent bathing of separated cellulose film for complete washing of chromophores. This approach could recover $66 \pm 1\%$ of the chromophores. The reclaimed chromophores were reused for crystallization in a fresh cellulose film to complete the circular loop, whereas clean gelatin and BC residues can be subjected to biodegradation. The recycled chromophores crystals entrapped in BC–G films showed upconversion emission. Our previous work reported recyclable red/far red to blue TTA-UC bioplastics by fabricating chromophores in a gelatin-TX-100-reduced co-assembled film[17]. Wherein we reported around 67% recovery of the chromophores in the TX-100-reduced phase which could be refabricated to form bioplastics. However, the surfactant TX-100-reduced is still toxic at high concentrations if discharged into the natural streams due to non-biodegradability[18]. Hence, we wanted to remove the surfactant phase[17] from the bioplastics system to make them more sustainable. We think that utilizing the pre-programmed TTA-UC crystals doped biopolymers offers a more sustainable approach, as it allows easy recycling of chromophores via physical separation and anti-solvent bathing at the EoL. Cellulose has been reported previously as matrix to host TTA-UC chromophores, specifically in the form of cross-linked cellulose microcapsules that are loaded with

TTA-UC solutions and subsequently coated with cellulose nanofibers[19]. On the contrary, this work introduces an innovative approach to employing cellulose nanofibers both as host and template for the crystallization of TTA-UC chromophores in their photo-functional state, a method that appears unprecedented to the best of our knowledge.

## Results and discussion
### Preparation and characterization of transparent photonics back-sheet bioplastic films

The BC films were prepared by culturing the *Komagaetaeibacter xylinus* (*K. xylinus*) bacterial strain. *K. xylinus* is a gram-negative bacterium that produces cellulose with a distinct nanofibrillar interwoven structure[20,21]. The details of the bacteria culture are provided in the methods section. A translucent dry film with a milky white appearance and an average thickness of 26 μm was obtained from the *K. xylinus* culture. (Fig. 2a, inset). Air-dried BC films appear translucent due to the light scattering induced by surface roughness, fiber coalescence during drying, and porosity[22]. Transmission electron microscopy image of the BC film confirmed long fibers and pore size varying from 50 to 500 nm (Fig. 2a). The P-XRD spectrum of the BC film confirmed its crystallinity as it shows typical peaks at $2\theta = 14.4°$, $16.8°$, and $22.7°$ corresponding to (100), (010), and (110) planes of cellulose $I_\alpha$ (Fig. 2c)[23]. The TGA curves confirmed the high thermal stability of the BC film with an onset of thermal degradation at 295 °C and inflection temperature at 332 °C measured from the first derivative with a mass change of −72% (Supplementary Fig. 1a). The thermal stability of BC film is competitive with polyvinyl chloride[24] and polyesters[25]. The BC should have optical transparency to qualify for optical applications as part of back-sheet material. But, as mentioned, BC internal random pores (Fig. 2a), large fiber size, and surface roughness cause large scattering due to the difference in refractive index between air ($n = 1$), and BC ($n = 1.41 \pm 0.01$) measured using ellipsometry (Supplementary Fig. 2a)[26]. Therefore, to increase the optical transparency, we have coated the BC film with another optically transparent biopolymer, gelatin type A (G) that has a refractive index ($n = 1.5 \pm 0.2$, Supplementary Fig. 2b)[27], as shown in Fig. 1a. The gelatin sol when cast on a BC film, fills up pores and smoothens the rough surface increasing the overall optical transparency of the film by adjusting the refractive index to $n = 1.48 \pm 0.25$ in the visible region (Supplementary Fig. 2c). The transmission electron microscopy image of BC–G film (Fig. 2b) confirmed the filling of the pores of BC films and smoothening of surface and the optical transparency increase (Fig. 2b, inset).

The P-XRD spectrum BC–G film (Fig. 2c) shows decrease in intensity of $2\theta$ peak of gelatin at 7.9° corresponding to the crystalline triplet helix of G and disappearance of $2\theta$ peaks at 14.4° and 16.8° of BC. The $2\theta = 22.7°$ of BC is retained in the broad amorphous region of G indicating physical interactions between G and BC. To qualify as photonics back-sheet material, BC–G films must also exhibit enough mechanical stability; this has been confirmed by dynamic mechanical analysis (DMA) (Supplementary Fig. 2a, b). The time sweeps DMA of the BC, and BC–G films show a high stiffness with a stable storage modulus ($E'$) of 2700 MPa, and 4150 MPa at 30 °C, over a time period of 15 min (Supplementary Fig. 3a, b). The mechanical properties of these films are comparable with that of synthetic plastics like polyvinylchloride ($E' = 650$ MPa at 30 °C), see Supplementary Fig. 4[28], and polyester ($E' \sim 2000$ MPa)[29]. The dielectric constant of photonics materials is the key property when considering effective exciton diffusion or migration. A high dielectric constant reduces the exciton binding energy and the charge carrier recombination losses, leading to improved performance[30]. We recorded the frequency dependence of the dielectric constant of both BC and BC–G films (Fig. 2d). The dielectric constant of films was calculated from the experimentally measured parallel-plate capacitance according to the standard equation[31,32].

$$\varepsilon_r = \frac{(C \times d)}{A \times \varepsilon_0}$$

where $\varepsilon_r$ is the relative permittivity or dielectric constant, $C$ is the measured capacitance, $d$ is the thickness of the film, $A$ is the capacitor's area, and $\varepsilon_0$ the permittivity of vacuum. Three BC films of varying thickness were prepared for the dielectric measurements. This was done by stacking wet BC films and air drying the stack. The stacking increased the dielectric constant of BC films from 2.32 to 3.98 at $10^6$ Hz which is consistent with the reported in the literature[31]. The variation between different films could arise due to the varying thickness and trapping of air between stacked films due to their porous nature. The dielectric constant of BC–G at $10^6$ Hz was found to be 4.86 which is higher than the back-sheet material of synthetic polymers (PVF/PVDF/PET) used in photonics industries (Table 1)[33]. Hence, the material is suitable for photonics applications.

## Crystallization of photonics chromophores in bioplastic back sheet

First, we attempted the in-situ integration of chromophores directly into the BC during the cellulose biosynthesis. For that, we mixed a water-soluble sensitizer–annihilator pair of Pt(II) meso-5,10,15,20-tetrakis-(*N*-methyl-4-pyridyl)porphine, and bis-sodium 9,10-diphenylanthracene-2,7-disulfonate in the bacterial feed solution. The bacteria survived in the presence of the chromophores and secreted BC. Even though the chromophores were entrapped into the BC film during in-situ secretion (observed from their fluorescence emission), a significant portion leached out during the NaOH washing of the BC film. Therefore, we opted to carry out ex-situ chromophore incorporation by their crystallization into water-soaked BC hydrogel films. The chromophores were crystallized on the biopolymer film by dropwise addition of a THF solution of DPA–PdTPBP into BC hydrogel, followed by air drying (Fig. 1b, see experimental section for detailed procedure). Crystallization of chromophores within the BC film was confirmed from bright and dark field polarized microscopy (Fig. 3a, b). The scanning electron microscopic (SEM) images also confirmed TTA-UC microplates with average dimensions $25 \times 10$ µm (Fig. 3c), embedded in the BC nanofibers (Fig. 3d). Ellipsometry measurements confirmed the change in refractive index from $1.41 \pm 0.01$ in BC film to $1.55 \pm 0.05$ in the BC–TTA-UC crystals film (Supplementary Fig. 2d). After confirming the co-crystallization of DPA–PdTPBP on BC film, we drop-casted a 20%

### Table 1 | The dielectric constant of the different polymers used in this work and reported[33]

| Sample | BC | Gelatin | BC–G | PV back sheet[33] |
|---|---|---|---|---|
| $\varepsilon_r$ (at $10^6$ Hz) | 2.32 | 28.42 | 4.86 | 2.45–2.77 |
| Thickness (µm) | 26 | 150 | 170 | – |

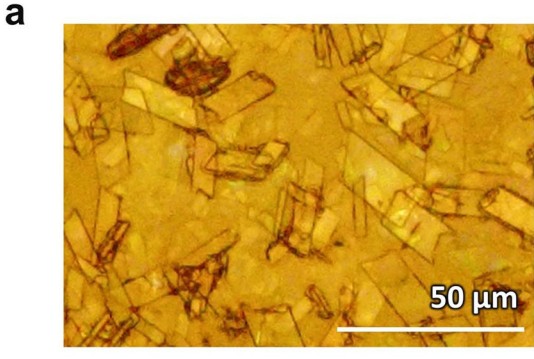

**a**

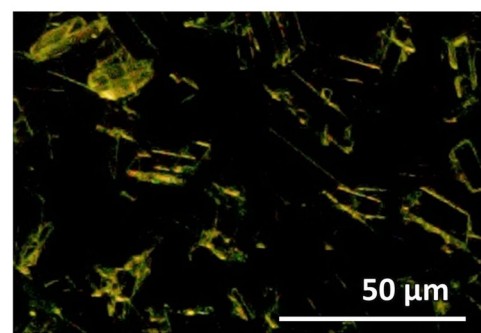

**b**

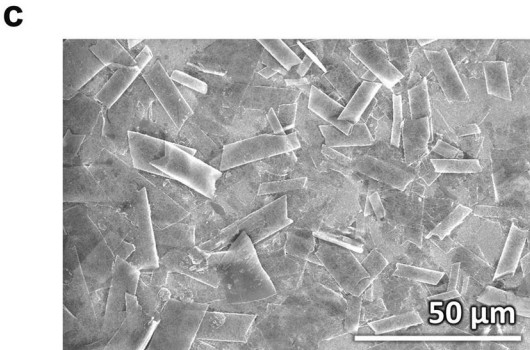

**c**

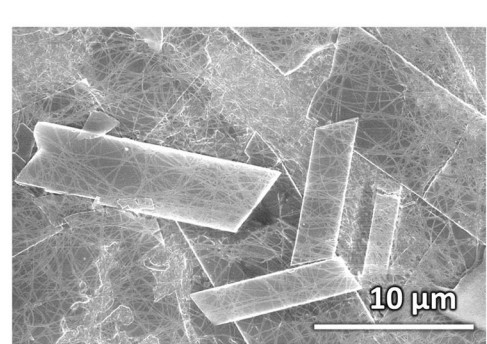

**d**

**Fig. 3 | Microscopic imaging of TTA-UC crystals embedded in BC film. a, b** Bright field and dark field polarized microscopic images and **c, d** SEM images of TTA-UC crystal microplates embedded in the cellulose nanofibers.

aqueous solution of gelatin on to it, which was then left to air dry overnight. Noteworthy is to mention here that despite of high concentration of gelatin in the BC–G composite film, the BC nanofibers play a key role in driving an ordered crystallization of TTA-UC chromophores into microplates (Fig. 3a, b).

This has been confirmed by doping TTA-UC chromophores into gelatin film alone resulting in circular aggregated chromophores crystals of varying shapes which does not show UC emission (Supplementary Fig. 5). The air-dried transparent BC–G–TTA-UC crystals film was subjected to photo-physical characterization (linear, and non-linear, see Fig. 4)[7,8,12].

## Linear and non-linear optical characterization of BC–G–TTA-UC crystals film

The linear optical emission in the film is indicated by fluorescence emission from the excited singlet state of the chromophores upon absorption of high energy photons (Fig. 4a–d) according to the following Eq. 1.

$$I_f = kI_o\Phi\left[1 - \left(10^{-\varepsilon bc}\right)\right] \qquad (1)$$

where $k$, $I_o$, $\Phi$, $\varepsilon$, $b$, $c$ denote proportionality constant, excitation intensity, fluorescence quantum yield, extinction coefficient, path length, and concentration of fluorophore, respectively.

The UV–vis spectrum of the BC–G–co-crystals film measured in the reflectance mode showed a broad absorption spectrum corresponding to the absorption maximum of DPA at 403 nm, and that of PdTPBP at 631 nm (Fig. 4b, black line). The fluorescence spectrum of DPA showed an emission maximum at 443 nm upon excitation with a 375 nm laser (Fig. 4b, blue line). The absorption and emission spectra of DPA crystals in BC and BG–G films were also measured separately to investigate the impact of gelatin coating. Nevertheless, no alteration in the spectra was observed (see Supplementary Fig. 6). The linear dependence on the excitation power density of the DPA crystals fluorescence was confirmed upon excitation with 375 nm CW laser with $f(x) = ax + b$ fit (Fig. 4c). The time-resolved fluorescence emission profiles of film samples 1, 2, and 3 (Supplementary Fig. 7b) show a tri-exponential decay for crystalline DPA systems with a significant reduction in fluorescence lifetime, $\tau_f$ (0.5–2.4 ns) of the first two fast decay components ($\tau_1$, and $\tau_2$) of DPA (Supplementary Table 1), compared to that in the organic solution of DPA ($\tau_f \approx 10$ ns)[34]. Also, the fluorescence quantum yield in the crystals estimated using integrating sphere[35] was found to be between $\Phi_f = 4.2$–4.5 % for samples 2 and 3 (Supplementary Fig. 7c, d) which is very low compared to an organic solution of DPA ($\Phi_f = 100\%$)[36,37]. The significant shortening of $\tau_f$, along with the reduction of $\Phi_f$ in the crystals, is likely due to non-radiative deactivation caused by impurities or other defects, and possibly due to back-FRET to the sensitizer PdTPBP. The fluorescence microscopic imaging of TTA-UC crystal plates shows bright blue emission typical of DPA upon excitation at 405 nm (Fig. 4d).

While cellulose-based linear optical materials have been reported for many applications[7,38] not much work has been done to exploit them for non-linear optics[7,19]. TTA-UC (Fig. 4e) is a non-linear optical process where the decay of annihilator triplets via TTA shows quadratic dependence on the delayed fluorescence (UC emission intensity) with time according to the following Eq. 2 [8,12,13].

$$I_{UC}(t) = \frac{1}{2}f\gamma_{TTA}\left[A^T(t)\right]^2 \qquad (2)$$

where $I_{UC}$ is UC emission intensity, $f$ is probability of singlet pair formation after TTA, the annihilators triplet decay rate, $\gamma_{TTA}$ is bimolecular annihilation constant, and $A^T$ is concentration of annihilator triplets.

Figure 4e shows an energy diagram of TTA-UC where absorption of a low-energy photon by the sensitizer is followed by a series of energy transfer steps (TET and TTA) operating via electron exchange (Dexter energy transfer) mechanism and eventually resulting in an upconverted excited singlet emission state of the annihilator. As discussed in the introduction section solid-state TTA-UC has broad applications in solar energy

harvesting via conversion and storage and optical anti-counterfeiting[12]. Hence, we studied the TTA-UC emission of DPA–PdTPBP crystals doped in BC–G film. For TTA-UC measurements three different films 1, 2, and 3 with different crystal concentrations were studied (Fig. 4f, see experimental section for sample preparation). All films showed TTA-UC emission with a maximum at 443 nm upon excitation at 633 nm CW laser in an aerated environment ($\Delta E_{\mathrm{anti-stokes}} = 0.83$ eV). The absence of phosphorescence emission of the sensitizer (PdTPBP) at $\approx800$ nm indicates high triplet energy transfer from the sensitizer to the annihilator. It is to be mentioned here that despite the triplet energy deficit of 220 meV of the PdTPBP ($T_1 = 1.55$ eV) compared to DPA ($T_1 = 1.77$ eV), TET proceeds efficiently (Fig. 4e). Therefore, TET in the crystals occur via endothermic route which is generally challenging as per the Boltzmann's law in thermal equilibrium[39,40]. For the same set of sensitizer–annihilator pair crystals, Li et al.[11] have reported thermally activated TET from temperature-dependent phosphorescence quenching which compensates for the 220 meV of the triplet energy deficit. Also, in crystals, the annihilator chromophores are closely packed in an ordered array; hence energy transfer is expected to operate via triplet energy migration (TEM).

As can be seen from Eq. 2, TTA-UC shows a two-stage dependence on the excitation power density. Stage one represents the quadratic regime in the low excitation power density region and stage two represents the linear regime in the high excitation power density range with higher annihilation rate. These triplets collide and undergo TTA to produce an emissive singlet state[12]. The $I_{th}$ is usually considered equal to 50% of the maximum UC emission quantum yield $\left(\Phi_{UC}^\infty\right)$[12], however, recently Murakami and Kamada[41] re-valuated this and clarified that this point corresponds only to 38.2% of the $\Phi_{UC}^\infty$. Figure 4f shows the quadratic to linear dependence of the UC emission intensity with an excitation power density of film 2. The $I_{th}$ corresponding to 38.2% and 50% of the $\Phi_{UC}^\infty$ have been calculated to be 1.82 W cm$^{-2}$, and 3.62 W cm$^{-2}$ (Fig. 4g). Figure 4h shows the digital image of blue upconverted emission by BC–G–DPA–PdTPBP crystal doped film upon 633 nm laser excitations in air. We note that cellulose-based TTA-UC systems were reported before but with a completely different design principle[19]. A cross-linked TTA-UC solution (perylene–PdTPBP) in microcapsules were embedded in the multi-layers of cellulose nanofiber matrix for oxygen protection[19]. However, TTA-UC was not stable upon continuous irradiation due to insufficient sealing by cellulose nanofibers against the oxygen diffusion which quenched the molecularly diffusing annihilator triplets in the microcapsules solution[38]. Remarkably, in the current system, compact packing of the crystals leave little space for oxygen entry and allow efficient migration of triplet among chromophores (Fig. 5). That is why TTA-UC is observed even in the naked crystals in the aerated environment (Supplementary Fig. 8a, b), and BC nanofibers in the current system have been used just to support the TTA-UC crystals as back sheet. Similarly, in our previous works we used gelatin as a bioplastic matrix to disperse liquid-surfactant containing TTA-UC chromophores (DPAS-PtOEP), and TIPS-Anthracene-PdTPBP) and as oxygen protector[16,42].

However, in this work gelatin has been used as a matrix to increase the optical transparency of the BC-loaded TTA-UC crystals, though it is also expected to provide oxygen sealing (Supplementary Fig. 9). We also notice that the TTA-UC quantum yield ($\Phi_{UC}$) in the current system could not be determined via the absolute method using integrating sphere. This is largely due to the low fluorescence quantum yield ($\Phi_f = 4.5\%$) of the prepared crystals. However, this study aims to demonstrate the feasibility of using a biopolymer back sheet supporting a crystalline non-linear optical system. In our future endeavors, we are committed to preparing crystals of high TTA-UC quantum yield with a better chromophore design.

## Recyclability of the TTA-UC chromophores and bioplastic materials via co-assembly–disassembly approach

The BC, TTA-UC crystals, and G collectively form a distinctive physical co-assembly (Figs. 1b and 3) that facilitates their easy disassembly through physical separation once their purpose has been served. The recycling process of the BC–G–DPA–PdTPBP crystal film is illustrated in Fig. 6.

 

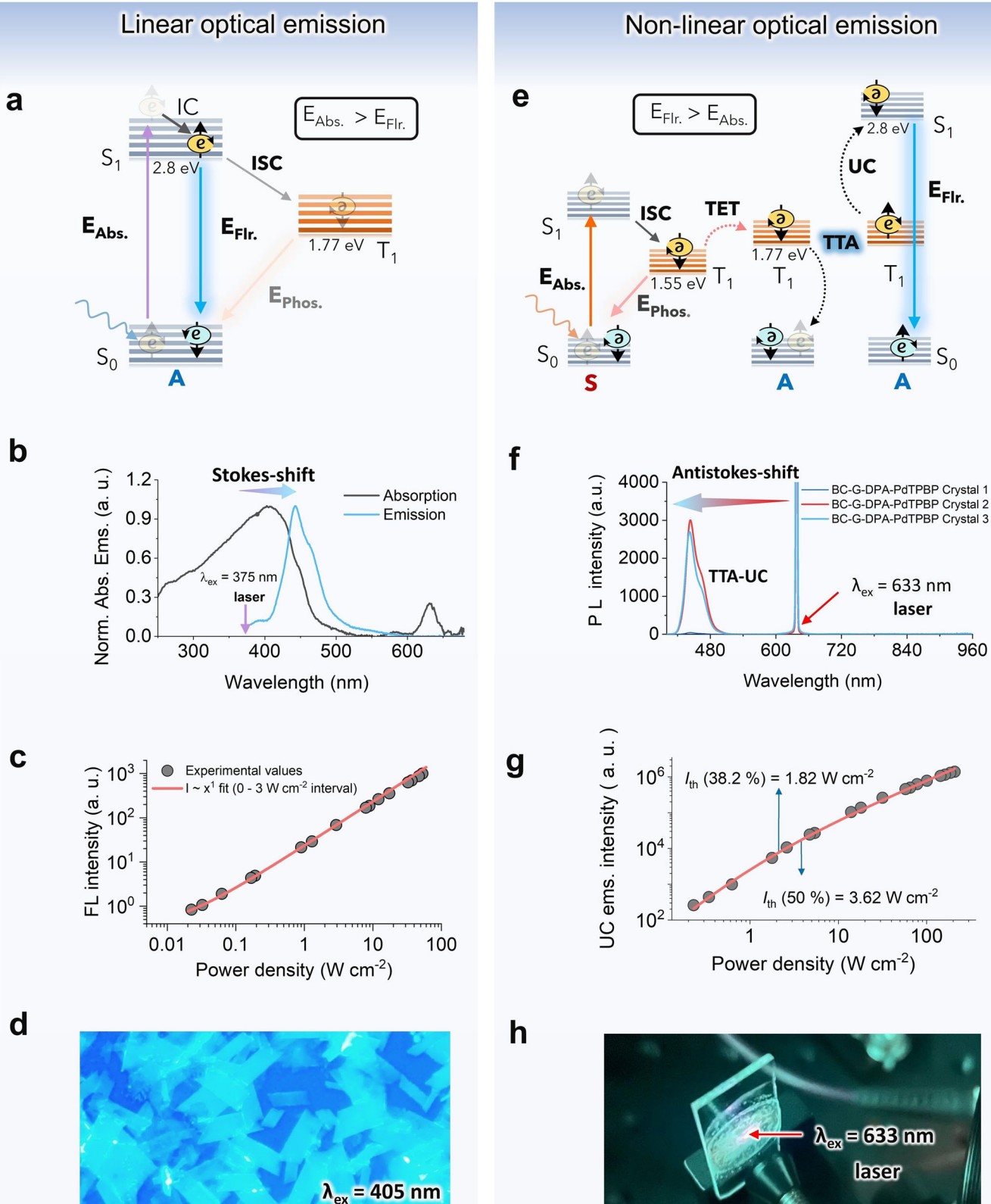

**Fig. 4 | Linear and non-linear optical properties of BC–G–TTA-UC crystal film.** **a** Energy diagram showing fluorescence emission from the excited singlet state, **b** normalized absorption, and steady-state emission spectra showing stokes-shift ($\lambda_{ex}$ = 375 nm laser), **c** power dependence of the fluorescence emission intensity, and **d** fluorescence microscopy image ($\lambda_{ex}$ = 405 nm). **e** Energy diagram showing a series of energy transfer steps in TTA-UC emission from the excited singlet state. **f** Steady-state UC emission spectra, **g** power dependence of the UC emission intensity, and **h** digital image of UC emission by DPA–PdTPBP crystals doped BC–G film in air ($\lambda_{ex}$ = 633 nm fiber laser, 10.2 W cm$^{-2}$, 532 nm S. P. filter).

**Fig. 5 | Energy transfer in DPA–PdTPBP crystals.**
Illustration of the energy transfer between sensitizer
(PdTPBP) doped in the annihilator (DPA) crystals.
TET triplet energy transfer, TEM triplet energy
migration, UC upconversion, S sensitizer, and A
annihilator.

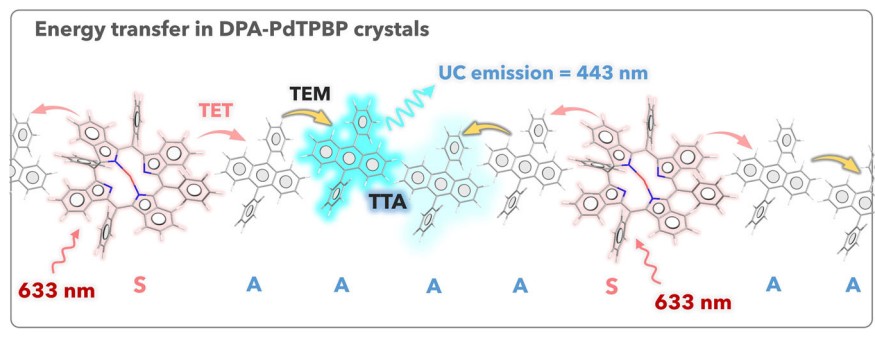

**Fig. 6 | Recycling of BC–G–TTA-UC crystals film.**
**a** Digital images of the recycling of BC–G–DPA–UC
crystal film via physical separation in water and anti-
solvent bathing with THF. **b** Upconversion emission
spectrum, and digital images of TTA-UC emission
in recycled BC–G–DPA–PdTPBP film upon exci-
tation with a 633 nm fiber laser at a power density of
10.2 W cm$^{-2}$.

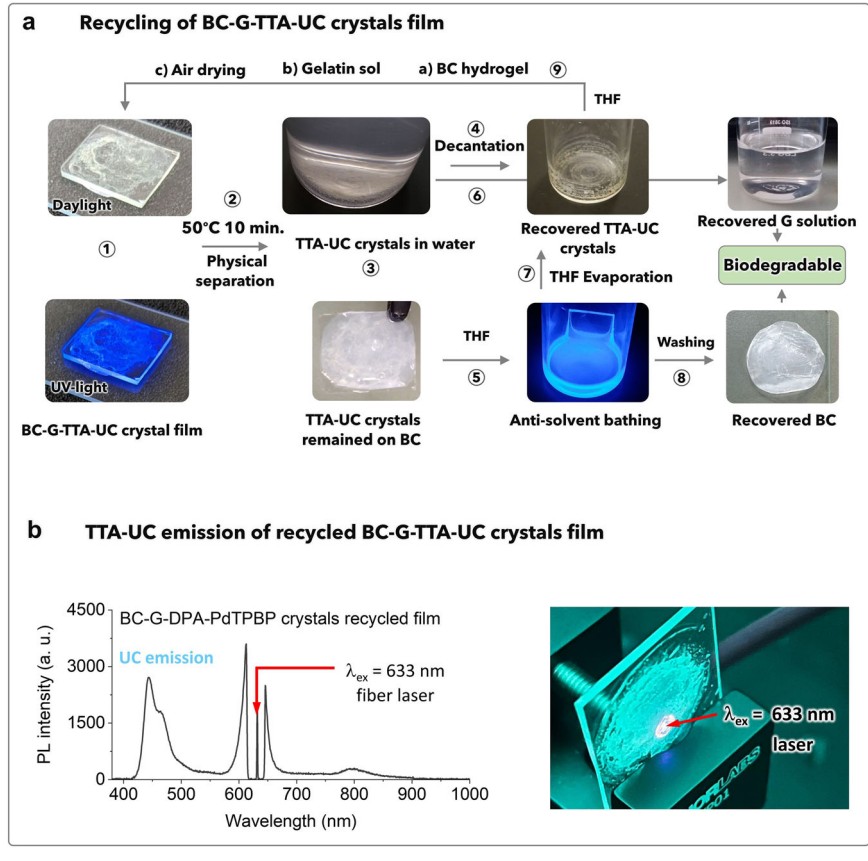

To initiate the disassembly, the BC–G–DPA–PdTPBP crystal film is first placed in hot water at 50 °C for 10 min, while being continuously agitated. At this temperature, gelatin dissolves in water, exposing the TTA-UC crystals. Subsequently, the BC film, which remains unaffected by the hot water is immersed in the THF solution for anti-solvent bathing. It results in the dissolution of TTA-UC chromophores in the THF. The chromophores crystals, separated from the gelatin aqueous solution, were then mixed with the THF solution of chromophores recovered from BC film. The THF is subsequently evaporated under reduced pressure, allowing reclaiming of the chromophores (Fig. 6a). Approximately 66 ± 1% of the DPA was success-fully recovered by combining both methods, as determined by absorption spectra analysis (Supplementary Fig. 10). The very low concentration of PdTPBP in comparison to DPA prevented the observation of its peak in the absorption spectra. The loss of the remaining 33 ± 1% of chromophores occurred during the cleaning process of the BC film with THF and partially during decantation. The recovered crystals were further washed with THF on the Whatman filter paper for the complete removal of organic residues of gelatin.

The purity of the recovered DPA was validated by comparing the ¹H-NMR spectrum of pure DPA, fresh TTA-UC crystals, and recycled TTA-UC crystals, where no change was observed in the chemical shift values of DPA (Supplementary Fig. 12). The clean milky white BC residues and gelatin aqueous solution obtained at the EoL can undergo biode-gradation due to their natural degradation mechanism. Polarized micro-scopy images of the BC film before and after crystal washing are shown in Supplementary Fig. 12. The digital images of the recovered biomass (BC residues and gelatin aqueous solution) under daylight and ultraviolet light demonstrate the absence of any chromophores (Supplementary Fig. 13). The recovered DPA chromophores were reused for crystallization on a fresh BC hydrogel film upon combining with a new PdTPBP solution to prepare a new TTA-UC film to complete the circular loop. The fluorescence micro-scopy image of the TTA-UC crystals fabricated on the BC film is shown in Supplementary Fig. 14. The photoluminescence spectrum of the recycled BC–G–TTA-UC film, along with a digital image illustrating TTA-UC emission upon excitation with a 633 nm fiber laser, 10.2 W cm$^{-2}$ are shown in Fig. 6b.

## Conclusions

In Summary, we demonstrate an eco-design for a solid-state co-assembly↔disassembly approach to prepare non-linear optically functional photonics bioplastics. The approach comprises four steps. (1) Preparation of optically transparent photonics back-sheet biopolymer film (BC–G composite) with suitable thermo-mechanical ($T_d > 300\ °C$, and $E' = 4150$ MPa), and dielectric ($\varepsilon = 4.85$ at $10^6$ Hz) properties competent with synthetic plastics. (2) Crystallization of functionally active photon upconversion chromophores into those bioplastics. (3) Non-linear optical emission via upconversion of red light ($\lambda_{ex} = 633$ nm) into blue light ($\lambda_{em} = 443$ nm) in the BC–G bioplastic film, and (4) easy recycling of $66 \pm 1\%$ of annihilator at the EoL via separation of chromophores crystals which can be reused. The approach proposes new design principles for the easy integration of photofunctional materials with biopolymers to address the critical situation of ever-increasing plastic waste from the plastic photonics industries. Our approach is based on boosting the circularity in the processing of photonic devices and contributing to the bioeconomy principles. Our future endeavors will be focused on further fine-tuning this approach with a broad scale of chromophores-bioplastic materials. We also noticed that the developed bioplastics back sheet has suitable thermo-mechanical properties for photonics, but water vapor permeability is still an issue; it will be among the key agenda of our future research.

## Methods

### Materials

*K. xylinus* strain was purchased from the Spanish type of culture collection (Coleccion Espanola de Cultivos Tipo (CECT)). Gelatin type A porcine skin 90–120 g bloom, and DPA (97%) were purchased from Sigma Aldrich. PdTPBP was purchased from Frontier Scientific. D-Glucose and citric acid were purchased from Fisher Chemical; peptone, yeast extract, and agar were Conda Lab supplies; and the sodium di-basic hydrogen phosphate was provided from Sigma Aldrich and used as received. All solvents were purchased from Carlo Erba Reagents and Sigma Aldrich.

### Microscopy

Optical and fluorescence microscopic images of DPA–PdTPBP crystals doped in BC film, at ×50 magnification, were acquired using an Olympus BX51 light polarized microscope (Olympus Co., Tokyo, Japan). For fluorescence imaging the samples were excited at 405 nm. Transmission electron microscopic images of BC and BC–G films were acquired using JEOL JEM1210 (120KV) transmission electron microscope with a point-to-point resolution of 3.2 Å. SEM images of DPA–PdTPBP crystals doped in BC film were acquired using Quanta 200 ESEM FEG from FEI.

### Thermogravimetric analysis (TGA)

TGA assays of BC and BC–G films were carried out with a NETZSCH -STA 449 F1 Jupiter, which allows for simultaneous TGA and differential scanning calorimetry/differential thermal analysis. Samples were analyzed with a heating rate of $10\ °C\ min^{-1}$ from 25 to 500 °C under $N_2$ flow.

### Dynamic mechanical analysis (DMA)

DMA was performed by using a Q800 machine (TA Instruments) working in tension mode. The specimens were cut using a ZCP 020 die cutter machine (ZwickRoell) to obtain rectangular shapes with a width of 4 mm and a length of 7.4 or 7.9 mm for the BC–G and BC materials, respectively. The thickness of the BC–G material was 170 μm and the thickness of the BC film, measured by using a MEGA-CHECK 5F-ST apparatus (NEURTEK S.A) was equal to 26 μm. The materials were held isothermally at 30 °C for 2 min. Each sample was then pre-loaded at a constant force (0.1 N for the BC–G and 0.05 N for the BC) and further deformed dynamically at an amplitude varying from 0.01 to 0.1% and at the frequency of 1 Hz. The storage modulus, $E'$, was measured as a function of the strain for various amplitudes and as a function of the time at the amplitude 0.05%.

### Dielectric measurements

The dielectric measurements were then done using Agilent E4980A Precision LCR meter. To measure the capacitance and dielectric loss, parallel-plate capacitors of the samples were fabricated by attaching the samples to a platinum-coated silicon substrate using silver paste and this acted as the bottom electrode. The top electrode was made using small droplets of silver paste that were put on top of the sample using a needle. The area of the top electrode was determined using optical microscopy.

### X-ray diffraction measurements

X-ray diffractograms of the BC and BC–G films were measured using Bruker. The D8 DISCOVER is an X-ray diffraction instrument equipped with four motorized axes stage, having as typical applications: XR reflectometry, rocking measurements, RSM measurements, and structural phase identification.

### UV–vis spectroscopy measurements

UV–vis measurements of the chromophores solution and solid sample were done using a UV–vis–NIR spectrophotometer, a Jasco V-780 with an operational range of 190–3300 nm. Solution state spectra were recorded in a quartz cuvette with a 1 cm path length. Solid-state measurements were carried out using a Diffuse Reflection Sphere (DRA-2500) accessory.

### Steady-state fluorescence spectroscopy measurements

The fluorescence spectra of the solid-state BC–G–TTA-UC crystals films were measured using AvaSpec-ULS2048CL-EVO—Avantes spectrometer. The sample film was excited with a 365 nm LED. The spectra were recorded with an average of 100 scans and an integration time of 10 ms. Fluorescence transients were measured by a time-correlated single photon counting system, PicoHarp 300 (Picoquant), by employing a pulsed semiconductor laser diode (repetition rate—1 MHz, pulse duration—70 ps) as an excitation source.

### Upconversion measurements

Photon upconversion (UC) measurements were carried out upon excitation with 633 nm using a 50 mW power continuous-wave semiconductor laser diode (Picoquant) and 633 nm fiber laser (80 mW). Steady-state UC emission spectra were measured using a back-thinned CCD spectrometer, PMA-12 (Hamamatsu), and Spectrometer Avantes Avaspec-ULS2048CL-EVO-RS-UA.

### Fluorescence (FL) and UC quantum yield measurements

FL and UC quantum yields were determined by utilizing an integrating sphere (Sphere Optics) coupled with the CCD spectrometer PMA-12 via an optical fiber and carrying out the procedure described by De Mello et al.[43] Fluorescence spectra were also measured using Spectrometer Avantes Avaspec-ULS2048CL-EVO-RS-UA.

### Ellipsometry measurements

Ellipsometry measurements were carried out using a Semilab Sopra GES5E VASE Ellipsometer, equipped with a rotating polarizer, collimated beam, or microspot, and spectral range of 230–1000 nm. Measurements were carried out at different angles of incidence varying from 55° to 75°. Pseudo optical constants from the ellipsometry measurement were derived by reversing the data determined at each angle of incidence"[44].

### Bacterial cellulose synthesis

BC was synthesized by the strain *K. xylinus* purchased from the Spanish type of culture collection (CECT). The medium for propagating bacteria was Hestrin–Schramm (HS) medium which consists of (g/L): D-glucose (20), peptone (5), yeast extract (5), sodium di-basic hydrogen phosphate (2.7), and citric acid (1.15). To produce BC films, sterile HS medium were disseminated in 24 wells plate. Afterward, we added the *K. xylinus* inocula, where the inocula represented 10% of the 2 ml overall volume. The plates were sealed and moved to be incubated at 30 °C for 5 days in the dark. Upon

the incubation, the BC films were collected and immersed in an aqueous solution of ethanol (50%) for 10 min, before they underwent the washing. The washing process is applied to get the BC films fully pure and eliminate the medium remnants and cell debris. The process includes immersing the BC films in boiling water twice, then in hot 0.1 N NaOH twice, 15 min each time. Afterward, the films were rinsed with excess dH$_2$O to restore the film's neutral pH. These neat BC films were then soaked in dH$_2$O and kept in the fridge at 4 °C until use.

### Preparation of the BC–G–DPA–PdTPBP crystals doped film
Into the 1 mL of 100 mM solution of DPA in THF added 10 μL of 100 μM solution of PdTPBP in THF making the annihilator to sensitizer ratio of 100,000:1. We first prepared the naked crystals by titrating against water according to the procedure of Li et al.[11] For the preparation of TTA-UC crystals doped BC film different volumes of DPA–PdTPBP solution in THF (50, 100, and 150 μL) added dropwise into the BC hydrogel. These samples are named sample 1, sample 2, and sample 3. The TTA-UC crystal-doped films were allowed to dry overnight. The crystallization of DPA–PdTPBP in the BC film was confirmed by SEM and polarized microscopy imaging. To increase the transparency of this film 250 μL of 20% gelatin type A solution was drop-casted onto the BC–TTA-UC crystalline film and allowed to dry overnight. The BC–G–DPA–PdTPBP crystal films were then subjected to photophysical, and TTA-UC properties analysis.

### Preparation of PVC film
The 1% PVC film has been prepared by dissolving 0.01 g of PVA in 1 ml of toluene at 50 °C. From the resulting solution, 260 μl was cast on a 3 × 1 cm glass plate. The film was dried overnight in the air and peeled off for DMA measurement. The thickness of the film was measured to be 0.11 mm.

### Recycling of BC–G–DPA–PdTPBP crystals doped film
In total, 3.2 mg TTA-UC crystals were dissolved in 100 μL of toluene. Five microliters of this solution was taken in a glass vial and diluted to 1 mL in spectroscopic grade toluene and named sample A. Sample A was subjected to UV–vis analysis to determine the concentration of chromophores (Supplementary Fig. 9a). The concentration of DPA in the solution was calculated to be 61.72 mM. Due to the very low concentration of PdTPBP compared to DPA, we could not observe its absorption spectra to calculate the concentration. The remaining 95 μL of the TTA-UC solution evaporated under reduced pressure and the dried residue was diluted with 95 μL of THF. This solution was doped into the BC hydrogel, followed by air drying and further coating with 20% gelatin sol, and air drying overnight. The film showed blue TTA-UC emission in the air (measured with 633 nm fiber laser excitation) and was named sample B. For recycling of the chromophores, sample B was immersed in aqueous solution at 50 °C upon continuous agitation. The gelatin dissolved in the hot solution, taking some of the TTA-UC crystals along, which settled at the bottom of the glass bottle. The remaining crystals remained on BC film which were collected via anti-solvent bathing in THF. The chromophores solution was used again for crystallization in BC hydrogel. The recovered BC was washed three times with THF for complete removal of the chromophores. The chromophore-free BC was confirmed by the absence of any fluorescence upon exposure to 365 nm LED. Both recovered BC and gelatin were discharged into the environment for biodegradation. The recycled DPA crystals were again dissolved in THF, followed by the addition of specific volume of THF solution of PdTPBP to get final concentration of 1 μM of PdTPBP. The recycled 100 μl of the TTA-UC solution was crystallized on a fresh BC hydrogel, followed by coating with gelatin to TTA-UC measurements.

### Data availability
The original datasets generated and analyzed during the current study are available from the corresponding author on reasonable request.

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

## Acknowledgements

This work was developed within the scope of the La-Caixa Junior Research Leadership-Post Doctoral Project PHOLCEB (ID: 100010434, fellowship code: LCF/BQ/P122/11910023). P.B. and M.M. acknowledge La-Caixa Foundation and the State Investigation Agency, through the Severo Ochoa Program for Centres of Excellence in R&D (CEX2023-001263-S) and Project PID2021-123873NB-I00 for financial support. K.M.P. acknowledges funding from the European Research Council (No. 101002131), the Swedish Energy Agency, the Göran Gustafsson Foundation, the Swedish Research Council, Swedish Research Council Formas, the European Research Council (ERC) under Grant Agreement CoG, PHOTHERM—101002131, the Catalan Institute of Advanced Studies (ICREA) and the European Union's Horizon 2020 Framework Program under Grant Agreement No. 951801. L.N. acknowledges Erasmus+ Traineeship Program. J.L. and K.K. acknowledge the "Universities' Excellence Initiative" program by the Ministry of Education, Science and Sports of the Republic of Lithuania under the agreement with the Research Council of Lithuania (Project No. S-A-UEI-23-6). A.R. acknowledges Grants PID2021-122645OB-I00 and CEX2019-000917 ("Severo Ochoa" Program for Center of Excellence in R&D) funded by MCIN/AEI/10.13039/501100011033 and by FEDER, "A way of making Europe", and The Generalitat de Catalunya with Grant No. 2021SGR00446. N.C. acknowledges the group eb-POLICOM/Polímers i Compòsits Ecològics i Biodegradables, a research group of the Generalitat de Catalunya (Grant 2021 SGR 01042). D.A.F. is grateful to the Egyptian Ministry of Higher Education and Scientific Research for his post-doctoral grant. G.D.L. acknowledges funding from the Grant RYC2021-032524-I funded by MCIN/AEI/10.13039/501100011033 and by "European Union NextGenerationEU"/PRTR. G.D.L and U.S. acknowledge Prof. G. Catalan for the use of the LCR meter.

## Author contributions

P.B. conceived the idea, designed, and performed the experiments (TTA-UC film preparation, polarized microscopy, photo-physical characterization, recycling), interpreted the data, and wrote the original version of the paper. K.M.P. conceived the idea, designed the work, interpreted the data, and contributed to the paper preparation. A.R. conceived the idea, interpreted the data, and contributed to manuscript preparation. D.A.F. cultured the bacteria and provided BC hydrogels and their characterization and revised the manuscript. M.M. performed SEM and TGA. L.N., P.B., J.L., and K.K. performed upconversion and time-resolved fluorescence measurements. G.D.L. and U.S. performed dielectric measurements and interpreted the data. G.D.L. also performed the ellipsometry measurements. N.C. performed DMA measurements and interpreted the data. All authors contributed during MS revision.

## Funding

## Competing interests
