## [Peer Review File · Communications Materials]

19th Mar 24

Dear Professor Moth-Poulsen,

Thank you for submitting your manuscript, "Towards Sustainable Photonics: Recyclable Photon Upconversion Crystals Doped Bacterial Cellulose Composite Films", to Communications Materials. It has now been seen by 2 referees, whose comments are appended below. In the light of their advice I regret to inform you that we cannot publish your manuscript in Communications Materials.

You will see that, while the reviewer 2 finds your work of interest, reviewer 1 raise substantive concerns that cast doubt on the advance your findings represent over related work and the strength of the novel conclusions that can be drawn at this stage. In particular, reviewer 1 suggests that there is insufficiency evidence to support claims in the manuscript, such as for the mechanical properties and general NMR data. Reviewer 2 also echoed that further experiments are required. Unfortunately, these reservations are sufficiently important to preclude publication of this study in Communications Materials.

I am sorry that we cannot be more positive on this occasion and thank you for the opportunity to consider your work.

Best regards,

Dr Jet-Sing Lee

Associate Editor

Communications Materials

Reviewers' comments:

Reviewer #1 (Remarks to the Author):

The manuscript entitled “Towards Sustainable Photonics: Recyclable Photon Upconversion Crystals Doped Bacterial Cellulose Composite Films” reported an eco-design for a solid-state assembly disassembly approach to prepare non-linear optically functional photonics bioplastics. However, the presented work only proposes a preliminary concept - non-linear optically functional photonics bioplastics - that lacks sufficient supporting data. While we appreciate the introduction of this design principle, the manuscript does not provide enough evidence to strongly support the proposed concept. Therefore, we cannot recommend this work for publication in Communication Materials. The other major concerns listed below should be emphasized:

1. In this manuscript, the choice of materials by the author is not innovative enough.
2. Line 191: To compare the mechanical properties of these films with those of synthetic plastics, it is necessary to provide experimental data or citations to support this statement.
3. Line 172: What is the thickness of the gelatin after it is poured? How can repeatability be ensured?
4. The introduction of luminescent groups into high-refractive-index polymer systems generally increases the refractive index of the material. Under this assumption, please compare the photorefractive indices of films before and after the introduction of TTA-UC chromophores crystals. Additionally, the manuscript does not compare the fluorescence quantum yield in the crystals and in the solutions.
5. Please provide complete $^1\text{H-NMR}$ spectra and data for pure DPA, fresh TTA-UC crystals, and recycled TTA-UC crystals.
6. Authors are requested to review the entire text for formatting errors. For example, there are two scale descriptions in both Fig. S7 and Fig. S3.

Reviewer #2 (Remarks to the Author):

In this study, the author explores the utilization of bioplastic materials for nonlinear optical applications, employing recyclable crystal-doped photonic bioplastics instead of synthetic plastics. By incorporating TTA-UC chromophore crystals into bacterial cellulose (BC) films, an upconversion effect from red to blue light was achieved. The study confirms the mechanical stability of the BC-G composite film as a photonics back-sheet material and verifies the presence of TTA-UC crystals in the bacterial cellulose film. Additionally, the linear and nonlinear optical properties of BC-G-DPA-PdTPBP doped film were investigated. Furthermore, a significant portion of the DPA could be recovered through physical separation or antisolvent immersion. This research introduces new avenues for designing recyclable optical bioplastic materials in energy harvesting

applications, contributing to the concept of a circular bioeconomy. Consequently, the manuscript holds potential for acceptance, contingent upon addressing the following points:

In Figure 3c-d, SEM images of TTA-UC crystal microplates embedded in the cellulose nanofibers, the scale size of the bar in Figure 3d is unclear, potentially misinterpreted as 50 μm instead of 10 μm .

Ensure consistency in the abbreviations used throughout the manuscript, aligning them with those mentioned in the abstract.

Investigate the difference in fluorescence lifetime of DPA in G-BC-DPA-PdTPBP film samples 2 and 3, as compared to sample 1, and provide an explanation for the observed discrepancy.

Conduct a detailed comparative study of the photophysical properties of sensitizer and acceptor in BC and BC-G films, including absorption, emission, and lifetime curves, to better illustrate the effect of gelatin addition.

Address the upconversion emission of the recovered chromophores after resampling, comparing it to the emission before recovery, as described in the last section of the main text.

Provide the error associated with the reported recovery rate of DPA (approximately 66.4%), and clarify how this value was obtained, potentially by conducting multiple experiments.

Reviewers' comments:

Reviewer #1 (Remarks to the Author):

Comment 1. The manuscript entitled “Towards Sustainable Photonics: Recyclable Photon Upconversion Crystals Doped Bacterial Cellulose Composite Films” reported an eco-design for a solid-state assembly disassembly approach to prepare non-linear optically functional photonics bioplastics. However, the presented work only proposes a preliminary concept - non-linear optically functional photonics bioplastics - that lacks sufficient supporting data. While we appreciate the introduction of this design principle, the manuscript does not provide enough evidence to strongly support the proposed concept. Therefore, we cannot recommend this work for publication in Communication Materials.

Answer: We agree with the reviewer that this is a preliminary concept, but we do not agree that we have not provided sufficient supporting data. We have provided enough details to support non-linear optics in the hybrid biopolymers film by doing a complete characterization of TTA-UC which is a non-linear optical process and is advantageous over many other processes like 2 photon absorption and excited state absorption, etc. due to operation at lower excitation intensities

The other major concerns listed below should be emphasized:

Comment 2. In this manuscript, the choice of materials by the author is not innovative enough.

Answer: Since this manuscript is based on exploring the potential of natural polymers materials can find potential uses in non-linear optics with regards to sustainability, we think we have used the correct biomaterials with which this concept could be established.

Comment 3. Line 191: To compare the mechanical properties of these films with those of synthetic plastics, it is necessary to provide experimental data or citations to support this statement.

Answer: According to reviewers suggestions, we have now provided DMA data of 1% polyvinyl chloride film (Fig. S4), and literature references of the DMA data of polyesters in the revised MS and SI, for a valid comparison.

Fig. S4. DMA profiles (Storage modulus vs time sweep profile) of 1% polyvinyl chloride film

Comment 4. Line 172: What is the thickness of the gelatin after it is poured? How can repeatability be ensured?

Answer: The thickness of gelatin in the film is 0.15 ± 0.01 mm. It can be easily controlled by adding a specific volume of hot gelation sol in a specific area. For example, we use 250 μ l of gelation sol on 1 cm^2 of bacterial cellulose-crystal film. Since the gelatin solution is viscous, it does not flow away while casting. We have also tested a 1 cm^2 glass rim as shown below to control the thickness. However, the direct casting method works the best and gives thickness with an error of ± 0.01 mm.

Comment 5. The introduction of luminescent groups into high-refractive-index polymer systems generally increases the refractive index of the material. Under this assumption, please compare the photorefractive indices of films before and after the introduction of TTA-UC chromophores crystals. Additionally, the manuscript does not compare the fluorescence quantum yield in the crystals and in the solutions.

Answer: As per the reviewer's suggestion, we have now measured the refractive indexes of the BC film before and after doping with crystals using the ellipsometry method. Additionally, the change in R. I. of BC film upon coating with the gelatin film is also provided in the revised Supporting Information as Fig. S2.

Fig. S2. Refractive index vs excitation energy profiles at different angle of incidence of a, bacterial cellulose b, gelatin c, gelatin-bacterial cellulose, and d, bacterial cellulose-TTA-UC crystals.

Additionally, the fluorescence quantum yield of DPA in the crystals and solution has been compared in the revised Manuscript with literature references.

Comment 6. Please provide complete $^1\text{H-NMR}$ spectra and data for pure DPA, fresh TTA-UC crystals, and recycled TTA-UC crystals.

Answer: We want to inform the reviewer that the $^1\text{H-NMR}$ spectra of pure DPA, fresh TTA-UC crystals and recycled TTA-UC crystals were already provided in the Supporting information as Fig.S6 in the previous submission. As can be seen from $^1\text{H-NMR}$ profiles, no difference in the chemical shifts was observed.

Comment 7 Authors are requested to review the entire text for formatting errors. For example, there are two scale descriptions in both Fig. S7 and Fig. S3.

Answer: We thank the reviewer for noticing this. But the scales in Figures S3 and S7 are the original, what we obtained during microscopic imaging. So, we cannot change them.

Answer to Reviewer #2 (Remarks to the Author):

Comment 1. In this study, the author explores the utilization of bioplastic materials for nonlinear optical applications, employing recyclable crystal-doped photonic bioplastics instead of synthetic plastics. By incorporating TTA-UC chromophore crystals into bacterial cellulose (BC) films, an upconversion effect from red to blue light was achieved. The study confirms the mechanical stability of the BC-G composite film as a photonics back-sheet material and verifies the presence of TTA-UC crystals in the bacterial cellulose film. Additionally, the linear and nonlinear optical properties of BC-G-DPA-PdTPBP doped film were investigated. Furthermore, a significant portion of the DPA could be recovered through physical separation or antisolvent immersion. This research introduces new avenues for designing recyclable optical bioplastic materials in energy harvesting applications, contributing to the concept of a circular bioeconomy. Consequently, the manuscript holds potential for acceptance, contingent upon addressing the following points:

Answer: We are thankful to the reviewer for positive comments about our work.

Comment 2. In Figure 3c-d, SEM images of TTA-UC crystal microplates embedded in the cellulose nanofibers, the scale size of the bar in Figure 3d is unclear, potentially misinterpreted as $50\ \mu\text{m}$ instead of $10\ \mu\text{m}$.

Answer: We thank the reviewer for pointing this out. However, we want to inform that these are zoom-out images, so the scale bar is increased accordingly. For better clarity, we have now provided the actual images in the revised supporting information and cited them in the main text. The original images are shown here for the reviewer's consideration.

Fig. S3. SEM images of DPA-PdTPBP Crystals doped in BC film; **a**, at scale bar of 50 μm , and **b**, at a scale bar of 10 μm .

Comment 2. Ensure consistency in the abbreviations used throughout the manuscript, aligning them with those mentioned in the abstract.

Answer: As per reviewer's concern, we have now corrected the abbreviations throughout the manuscript.

Comment 3. Investigate the difference in fluorescence lifetime of DPA in G-BC-DPA-PdTPBP film samples 2 and 3, as compared to sample 1, and provide an explanation for the observed discrepancy.

Answer: As per the suggestions of the reviewer, we have now also recorded the fluorescence lifetime data of all films and provided an explanation for this discrepancy.

Comment 4. Conduct a detailed comparative study of the photophysical properties of sensitizer and acceptor in BC and BC-G films, including absorption, emission, and lifetime curves, to better illustrate the effect of gelatin addition.

Answer: As per reviewers suggestion we have now carried out the absorption of emission characterization of DPA in BC and BC-G films and provided as Fig. SX in the revised Supporting Information as shown below. Since PdTPBP alone in these films due to low concentrations used, we could not measure its absorption and emission in these films.

Comment 5: Address the upconversion emission of the recovered chromophores after resampling, comparing it to the emission before recovery, as described in the last section of the main text.

Answer: As per the reviewer's suggestion, we have now provided an upconversion emission plot of the recovered DPA crystals. The plot is provided as Fig. 6b (also shown below) in the revised manuscript.

b TTA-UC emission of recycled G-BC-TTA-UC Crystals film

Comment: Provide the error associated with the reported recovery rate of DPA (approximately $66.4 \pm 1\%$), and clarify how this value was obtained, potentially by conducting multiple experiments.

Answer: As per reviewers' suggestion we have provided errors. We did it for three different samples (shown below), followed by absorption measurements of the recovered crystals to arrive at these values.

27th Aug 24

Dear Professor Moth-Poulsen,

Your manuscript titled "Towards Sustainable Photonics: Recyclable Photon Upconversion Crystals Doped Bacterial Cellulose Composite Films" has now been seen again by our referees, whose comments appear below. Reviewer 2 was not available to re-review this paper, but an additional reviewer, Reviewer 3, thinks your responses to them were appropriate. In light of their advice I am delighted to say that we are happy, in principle, to publish a suitably revised version in Communications Materials.

We therefore invite you to revise your paper one last time to address the remaining concerns of Reviewer 1. At the same time we ask that you edit your manuscript to comply with our journal policies and formatting style in order to maximise the accessibility and therefore the impact of your work.

EDITORIAL REQUESTS

* Your manuscript should comply with our policies and format requirements, detailed in our style and formatting guide (<https://www.nature.com/documents/commsj-phys-style-formatting-guide-accept.pdf>).

* Please edit your manuscript according to the editorial requests in the attached table, and outline revisions made in the right hand column. If you have any questions or concerns about any of our requests, please do not hesitate to contact me. It is important that each request be addressed in order to avoid delays in accepting your manuscript. Please upload the completed table with your manuscript files as a Related Manuscript file.

* The editorial requests table also includes a full list of the files that must be provided upon resubmission. Please upload your files according to this table.

* An updated editorial policy checklist that verifies compliance with all required editorial policies must be completed and uploaded with the revised manuscript. All points on the policy checklist must be addressed; if needed, please revise your manuscript in response to these points. Please note that this form is a dynamic 'smart pdf' and must therefore be downloaded and completed in Adobe Reader. Clicking this link will download a zip file containing the pdf.

OPEN ACCESS

Communications Materials is a fully open access journal. Articles are made freely accessible on publication. For further information about article processing charges, open access funding, and advice and support from Nature Research, please visit <https://www.nature.com/commsmat/open-access>

Please use the following link to submit your revised files:

[link redacted]

We hope to hear from you within two weeks; please let us know if the process may take longer.

Best regards,

Dr Jet-Sing Lee

Associate Editor

Communications Materials

orcid.org/0000-0002-6740-8700

REVIEWERS' COMMENTS:

Reviewer #1 (Remarks to the Author):

Comments:

In this paper, Moth-Poulsen et al. presented an innovative methodology for the incorporation of bioplastics with recyclable photonic crystals in photonic device applications and explored non-linear optical applications for biopolymers, specifically through the triplet-triplet annihilation photon upconversion (TTA-UC) process. The use of bacterial cellulose film (BC) as a substrate for the growth of TTA-UC crystals is a commendable solution. However, major revision is required before further consideration of this work, see comments below.

1. The novelty of this work should be further addressed in the Abstract and Introduction part.
2. The fonts and formats of some figures and captions need to be unified.
3. Please keep the horizontal and vertical axes of the pictures in Figure S11 consistent.
4. Please evaluate the necessity of presenting both Figure 3c-d and Figure S5a-b. If they are similar, consider retaining only one for clarity and to prevent confusion regarding the reuse of figures.
5. Please provide detailed information regarding the laser intensity used for the Upconversion emission spectrum throughout the manuscript.
6. The authors are suggested to improve the manuscript carefully to correct any grammar and format grammatical and formatting errors within the manuscript, for example ml, μ l, 630 laser et.al.
7. Please check the listed references carefully and to be properly cited.

Reviewer #3 (Remarks to the Author):

The authors have answered the comments properly in the revised manuscript and no further comments.